# Effects of Probiotics on Growth and Immunity of Piglets

**DOI:** 10.3390/ani12141786

**Published:** 2022-07-12

**Authors:** Ahmad Farid Nikmal Azizi, Ryoko Uemura, Mariko Omori, Masuo Sueyoshi, Masahiro Yasuda

**Affiliations:** 1Graduate School of Medicine and Veterinary Medicine, University of Miyazaki, Miyazaki 889-2192, Japan; fareedazizy@yahoo.com (A.F.N.A.); a0d802u@cc.miyazaki-u.ac.jp (M.S.); 2Laboratory of Veterinary Anatomy, Faculty of Agriculture, University of Miyazaki, Miyazaki 889-2192, Japan; 3Laboratory of Animal Health, Faculty of Agriculture, University of Miyazaki, Miyazaki 889-2192, Japan; uemurary@cc.miyazaki-u.ac.jp (R.U.); mrk.moo.0921@ezweb.ne.jp (M.O.); 4Center for Animal Disease Control, University of Miyazaki, Miyazaki 889-2192, Japan

**Keywords:** immunity, lymphocyte subsets, phagocytosis, piglets, probiotics

## Abstract

**Simple Summary:**

We evaluated effects of probiotics on growth and immune status of piglets. Body weights of probiotic-fed piglets were heavier than those of control piglets (*p* < 0.05). Relative populations of CD4^+^ and IgM^+^ cells isolated from the liver were increased (*p* < 0.05 and *p* < 0.01, respectively) in probiotic-fed piglets compared to control piglets. CD4^+^CD8^+^ T cells were decreased (*p* < 0.05) in jejunal Peyer’s patches of treated piglets. Phagocytosis of MHC class II^+^ cells isolated from the liver of probiotic-fed piglets was higher (*p* < 0.05) than that of control piglets. The probiotics have beneficial effects on the growth and health of piglets and could be good replacement for growth promoting antibiotics.

**Abstract:**

Growth promoting antibiotics are used in modern animal husbandry to promote growth and avoid infections. Negative effects of these antibiotics on human health are a big concern and they need to be replaced. Probiotics are expected to be a good replacement for growth promoting antibiotics. In this study, we evaluated effects of probiotics on growth and immune status of liver and secondary lymphoid organs of piglets. Body weights of probiotic-fed piglets were heavier than those of control piglets (*p* < 0.05) at days 30 and 45 of the experiment. Relative populations of CD4^+^ and IgM^+^ cells isolated from the liver were significantly increased (*p* < 0.05 and *p* < 0.01, respectively) in probiotic-fed piglets compared to control piglets. CD4^+^CD8^+^ T cells were significantly decreased (*p* < 0.05) in jejunal Peyer’s patches of treated piglets. Phagocytosis of MHC class II^+^ cells isolated from the liver of probiotic-fed piglets was significantly higher (*p* < 0.05) than that of control piglets. Phagocytosis of granulocytes isolated from the liver and peripheral blood of probiotic-fed piglets were also higher than those of control piglets. These results indicate excellent effects on growth and immune status of piglets. In conclusion, probiotics have beneficial effects on the growth and health of piglets and could be good replacement for growth promoting antibiotics.

## 1. Introduction

In response to increasing demands for food, and reduction in pathogenic bacteria, antibiotics are used more frequently in swine industries than in the past [1]. However, use of antibiotics as feed additives in livestock industries poses a high risk to human health and the generation of antibiotic resistance bacteria [2,3,4]. Thus, growth promoting antibiotics need to be replaced by better feed additives. 

Probiotics are defined as single or mixed cultures of live nonpathogenic microbes that have beneficial effects when administered in adequate numbers as a feed supplement [5,6]. Probiotic bacteria can alter the composition and activity of the gut microbiota, modulate the inflammatory response and improve the nonspecific intestinal barrier [7,8,9,10]. Research, including our laboratory, have shown that probiotics can activate immunity and prevent or reverse several pathological conditions in domestic animals [11,12,13,14]. The immunostimulatory effects of probiotics have also been observed in mice, where a significant increase in gut innate immune cells such as macrophages and dendritic cells was observed following oral administration of *Lactobacillus casei* CRL 43 [15]. Furthermore, probiotics can affect the host gut immune system by enhancing cell-mediated immunity, antibody production and stimulated T cell migration [16]. 

In addition, liver is a multifunctional digestive organ that makes bile, produces blood plasma protein and both stores and releases glucose. The organ contains numerous resident immune cells, including T cells, B cells, natural killer cells, dendritic cells and macrophages [17]. Because of the blood flow from the gut to the liver, the liver represents an important immune organ that is continuously faced with, and stimulated by, foreign antigens and bacterial products [17,18]. Therefore, liver immunity plays an important role in the clearance of gut derived antigens and protection of other organs from these foreign products. In rats, orally administrated lysozyme-modified probiotic components LzMPC (treating *Lactobacillus* sp., with lysozyme) were engulfed by liver macrophages [19]. Furthermore, the macrophages demonstrated enhanced cytokine production in response to lipopolysaccharide or LzMPC stimulation, ex vivo. However, the immunostimulatory and immunomodulatory effects of probiotics on the liver immunity of domestic animals remains unclear. Therefore, the aim of this study was evaluation of the effects of probiotics on growth and immune status of liver and secondary lymphoid organ in piglets.

## 2. Materials and Methods

### 2.1. Animals and Specimen Acquisition

Seventeen crossbred LWD (crossbred female LW (female Landrace and male Large Yorkshire) crossed with male Duroc) piglets weaned at the age of 25 days were bought from a commercial farm and housed in a windowless pigpen at the university. The pigpen was disinfected thoroughly before introduction of the pigs and high biosecurity was maintained throughout the experiment to avoid invasion of pathogens from outside. Piglets were divided into two groups, control group (*n* = 7) and treated group (*n* = 10). After a 7-day acclimatization period with ad libitum feeding and access to drinking water, probiotic (BIO-THREE Plus (BT), TOA Pharmaceutical CO., Ltd., Tokyo, Japan; containing *Streptococcus faecalis* strain T-110 (1 × 10^9^ CFU/g)*, Clostridium butyricum* strain TO-A (1 × 10^8^ CFU/g), and *Bacillus mesentericus* strain TO-A (1 × 10^8^ CFU/g)) was added to the diet for 60 days in the treated group. Both groups were fed “piglet Hatsuratsu” (Scientific Feed Laboratory Co., Ltd., Tokyo, Japan) at the beginning and gradually changed to “Miniature Swine Pellet” (Nosan Co., Ltd., Yokohama, Japan) after the age of 50 days. Both groups were fed twice a day at 12 h intervals and BT was administered orally to the treated group (4.0 g/day/head) before feeding. Body weights of both groups were measured at day 0 (beginning of the experiment), day 15, day 30, day 45 and day 60 (end of experimental period) of the experiment. At the end of the exposure period, all piglets were sacrificed by an intramuscular injection of mafoprazine mesylate (approximately 0.3–0.5 mg/kg, 1% Mafropan^®^**,** DS Pharma Animal Health Co. Ltd., Osaka, Japan), an intravenous injection of sodium pentobarbital (10 mg/kg, Somnopentyl^®^, Kyoritsu Seiyaku Co. Ltd., Tokyo, Japan), and an electric shock to induce cardiac arrest. All animal procedures were approved by the Institutional Animal Care and Use Committee of the University of Miyazaki (Approval No. 2019-001). Following sacrifice, small pieces of tissue from liver, soft palate tonsil (SPT), mesenteric lymph node (MLN) and jejunal Peyer’s patch (JPP) were stored. Single-cell suspensions were made by mincing in Hank’s balanced salts solution (HBSS, Sigma-Aldrich, St. Louis, MO, USA) and removing debris by a cell-strainer. Red blood cells were lysed by NH_4_Cl buffer. 

### 2.2. Lymphocyte Subsets Analysis

Subsets analysis was performed as described in our previous study [20]. Briefly, the cells were washed in PBS containing 0.05% sodium azide and 0.5% bovine serum albumin (BSA-PBS). These cells (1 × 10^5^ to 1 × 10^6^) were incubated with fluorescence labeled monoclonal antibodies (mAbs) as below for 1hr at 4 °C. After washing with BSA-PBS, the cells were suspended in BSA-PBS containing propidium iodide (1 µg/mL, Sigma Aldrich). The relative fluorescence intensities were examined by multicolor flowcytometry (FACS Canto^TM^ II system, Becton Dickinson, Franklin Lakes, NJ, USA). Fluorescence labeled anti-MHC class II (×200 dilution, TH81A5, Monoclonal Antibody Center at Washington State University, Pullman, WA, USA), anti-IgM (×100 dilution, PIG45A2, Monoclonal Antibody Center at Washington State University), anti-CD4 (×200 dilution, PT90A, Monoclonal Antibody Center at Washington State University), anti-CD8 (×200 dilution, PT36B, Monoclonal Antibody Center at Washington State University) and anti-γδ (×100 dilution, PGBL22A, Monoclonal Antibody Center at Washington State University) were used. For fluorescent labeling, HiLyte^TM^ Fluor 647 labeling kit-NH_2_ (Dojindo Laboratories, Kumamoto, Japan), HiLyte^TM^ Fluor 555 labeling kit-NH_2_ (Dojindo Laboratories) and FITC labeling kit-NH_2_ (Dojindo Laboratories) were used according to manufacturer’s instruction.

### 2.3. Phagocytosis Assay

Phagocytosis assay was performed as described in our previous study [14,21]. In brief, small pieces of liver were removed and a cell suspension was made by mincing in HBSS (Sigma-Aldrich) and removing debris with a cell strainer. Peripheral blood (PB) was collected in a heparinized tube. The buffy coat was collected after centrifugation (780 g for 10 min) at room temperature. The cells were washed with PBS and erythrocytes were lysed by NH_4_Cl buffer. After washing with PBS, the cells were suspended in RPMI1640 (Wako Pure Chemical Industries, Ltd., Osaka, Japan) containing 10% heat-inactivated fetal calf serum and antibiotics. Equal numbers of cells (4.0 × 10^6^ cells/mL for white blood cells, 2.0 × 10^6^ cells/mL for cells in liver) were then aliquoted into 24-well plates. One µL of a 2.5% FITC-latex bead (1 µm diameter, L1030, Sigma-Aldrich) was added and incubated for 1hr at 37 °C. After incubation, the cells were washed with ice cold 1 mM EDTA-PBS. And then, the cells were incubated with fluorescence labeled anti-granulocyte (PG68A, Monoclonal Antibody Center at Washington State University) or anti-MHC class II (TH81A5, Monoclonal Antibody Center at Washington State University) mAbs and then examined by multicolor flowcytometry (FACS Canto^TM^ II system).

The Phagocytosis index was measured as the percentage of FITC^+^TH81A5^+^ cells (phagocytizing latex beads) to total TH81A5^+^ cells, or the percentage of FITC^+^PG68A^+^ cells (phagocytizing latex beads) to total PG68A^+^ cells.

### 2.4. Statistical Analysis

Data were analyzed using the statistical software package SPSS for Windows (Version 20.0, SPSS Inc., Chicago, IL, USA). The Mann–Whitney U test was used to determine significant differences between the experimental groups. Results are expressed as the mean ± SD. *p*-values less than 0.05 were regarded as statistically significant.

## 3. Results

The health conditions of all piglets were good throughout the experiment. There were no symptoms of any disease in either group. Body weights of piglets in the BT treated group were heavier than those of the control group. Significant changes to the body weight were observed at days 30 and 45 of the experiment (*p* < 0.05) (Figure 1). 

Lymphocytes subsets analysis showed significantly higher relative populations of CD4^+^ cells (*p* < 0.05) and IgM^+^ cells (*p <* 0.01) in the liver of the BT treated group (Figure 2). Effects of probiotics on the lymphocyte’s population of MLN, JPP and SPT are shown in Table 1. The relative population of CD4^+^ T cells was slightly increased in these secondary lymphoid organs of the BT treated group compared to the control group, but the changes were not significant. MLN and SPT of the BT treated group showed slightly higher population of CD8^+^ cells compared to the control group. The percentage of CD4^+^CD8^+^ T cells relative population in the JPP of the BT treated group was significantly decreased (*p* < 0.05) as shown in Table 1. 

The phagocytosis indexes of granulocytes in PB and livers of the BT treated group were slightly higher than those of the control group (left side in Figure 3). In addition, the phagocytosis index of MHC class II^+^ cells in the liver of the BT treated group was higher (*p* < 0.05) than that of control group (right side in Figure 3). 

## 4. Discussion

Probiotics were observed to increase the numbers and phagocytic activity of monocytes and neutrophils in human peripheral blood [22]. They have been reported to increase the number of intestinal macrophages and to amplify production of IgM, IgG and IgA in probiotics-fed chicks [23,24]. Similarly, probiotic consumption in humans was associated with increased stimulation and activation of multiple immune cell populations, along with increased cytokine production [13,22]. Therefore, these studies reported that probiotics affect both innate and acquired immunity in humans and chicks [13,14,22,23,24]. This study evaluated the effects of probiotics on growth and immune status of liver and secondary lymphoid organ in piglets. The body weight of BT treated piglets was significantly increased (*p* < 0.05) at days 30 and 45 of the experiment. The daily body weight gain in both groups was not significantly different in this study (data not shown). Therefore, growth promotion by probiotics can be a good reason for considering probiotics as a replacement for growth promoting antibiotics. The limitation of this study was a lack of evaluation of food consumption and the food conversion ratio in BT-fed piglets. However, in our previous study, growth promotion of probiotics to BT-fed chicks has been reported [14]. The body weight in BT-fed chicks was significantly increased in the study, but daily body weight gain and food intake did not significantly change between control and BT-fed chick groups. 

The significantly higher relative populations of CD4^+^ cells (*p* < 0.05) and IgM^+^ cells (*p* < 0.01) in the liver of BT treated piglets was observed. In addition, phagocytosis of MHC class II^+^ cells in the liver of BT treated piglets was significantly increased (*p* < 0.05). Therefore, oral administration of BT promoted both innate and acquired immunity in the liver of piglets. The liver is an immune organ that is continuously confronted by foreign pathogens and commensal bacterial products [17]. Consistent with this immune challenge, many populations of immune cells reside in the liver, and approximately 25% of non-parenchymal liver cells are intra-hepatic lymphocytes [25]. Immune cells in the liver are responsible for the regulation of both an enhanced response to, and tolerance of, gut-derived antigens [26]. In addition, resident hepatic monocytes can secret numerous profibrotic, inflammomodulatory cytokines and growth factors [27]. Hepatic macrophages control the early phase of liver inflammation by secreting potent mediators, thereby playing important roles in the innate immune response [25]. Further functional analyses of hepatic lymphoid subsets could improve understanding of the probiotic immunomodulatory effect in animals. The population of CD4^+^CD8^+^ T cells in JPP was significantly lower (*p* < 0.05) in the BT treated group. The number of CD4^+^CD8^+^ T cells is very high in pigs and this population comprises approximately 8–64% of circulating T-cells. The CD4^+^CD8^+^ cell population consists predominantly of MHC class II^+^ restricted memory CD4^+^ cells which have acquired the ability to express the α-chain of CD8 [28,29,30]. Therefore, the production of CD4^+^CD8^+^cells in pigs is a post-thymic event. This study’s observation of a decrease in the population of CD4^+^CD8^+^ cells in JPP may account for the immunomodulatory effects of the probiotic. However, further investigation is needed to elucidate the significance of a CD4^+^CD8^+^ T cell population decrease in JPP. 

As an essential component of the innate immune response, phagocytosis is used for both antigen clearance and activation of the inflammatory response prior to antibody production [23,24]. In the previous study, we have described how probiotics improved chicken innate and acquire immunity [14]. In this study, we found significantly increased phagocytosis of MHC class II^+^ cells in the liver of BT-fed piglets. In addition, the phagocytosis of granulocytes in PB and liver were also increased in BT-fed piglets. Relative populations of CD4^+^ and IgM^+^ cells isolated from the liver were significantly increased. These results supported the idea that dietary supplementation with BT enhances innate and acquired immunity in the liver of piglets. 

## 5. Conclusions

The dietary supplementation with BT improved the growth of piglets, and the innate and acquired immunity in the liver of piglets. The phagocytosis of MHC class II^+^ cells and the populations of CD4^+^ cells and IgM^+^ cells in the liver were significantly increased. The probiotics have beneficial effects on the growth and health of piglets and could be a good replacement for growth promoting antibiotics.

## Figures and Tables

**Figure 1 animals-12-01786-f001:**
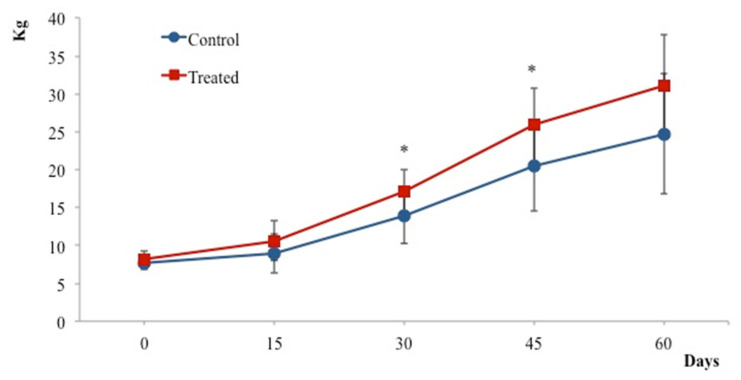
Effects of probiotic on body weight. The BT treated group shows a gradual increase in body weight following the experiment. Increase in body weight is significant 30 and 45 days following treatment. Values represent the mean ± SD. * *p* < 0.05 compared to controls.

**Figure 2 animals-12-01786-f002:**
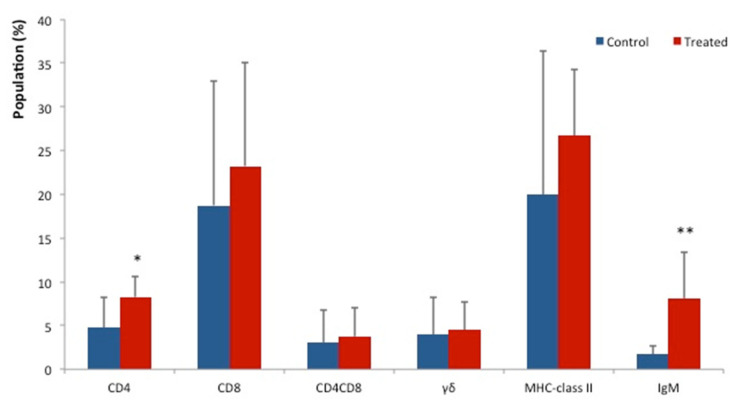
Immunostimulatory effects on the liver of BT-fed piglets. The populations of CD4^+^ cells and IgM^+^ cells in BT-fed piglets (*n* = 10) were significantly higher than those of control piglets (*n* = 7). The populations of CD8^+^, CD4+CD8^+^, γδ^+^ and MHC class II^+^ cells isolated from the livers of BT-fed piglets were also increased. Values represent the mean ± SD. * *p* < 0.05, ** *p* < 0.01, compared to controls.

**Figure 3 animals-12-01786-f003:**
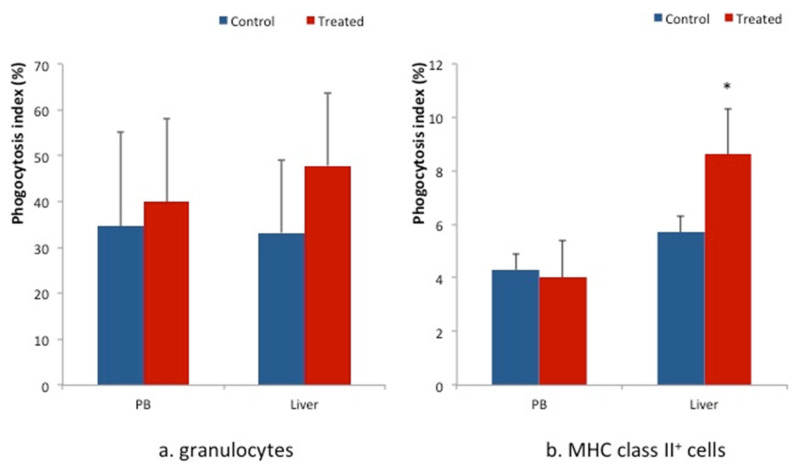
Phagocytosis indices of granulocytes (left side) and MHC class II^+^ cells (right side) isolated from the liver and PB. The Phagocytosis index of MHC class II^+^ cells in the livers of BT-fed piglets (*n* = 10) was significantly higher than that of control piglets (*n* = 7). The values represent the mean ± SD. * *p* < 0.05 compared to controls.

**Table 1 animals-12-01786-t001:** Relative population of T cells and B cells in secondary lymphoid organs of BT treated and control groups.

Subsets	Groups	MLN		JPP		SPT	
		Mean ± SD	*p*-Value	Mean ± SD	*p*-Value	Mean ± SD	*p*-Value
CD4	Control	27.00 ± 5.45	0.29	11.00 ± 3.76	0.5	11.40 ± 2.66	0.28
	Treated	30.00 ± 6.28		12.01 ± 3.21		14.00 ± 2.62	
CD8	Control	16.41 ± 3.65	0.72	13.00 ± 7.31	0.95	4.23 ± 0.76	0.57
	Treated	20.00 ± 3.44		13.00 ± 6.89		5.00 ± 1.16	
CD4CD8	Control	4.07 ± 1.60	0.94	7.20 ± 3.77	0.01	6.60 ± 1.28	0.5
	Treated	4.02 ± 1.54		4.30 ± 1.95		7.36 ± 1.75	
γδ	Control	10.34 ± 11.00	0.73	9.00 ± 6.14	0.5	18.13 ± 9.62	0.82
	Treated	9.00 ± 8.00		6.00 ± 10.4		17.00 ± 5.61	
MHC class II	Control	31.01 ± 6.14	0.09	41.00 ± 13.25	0.96	55.00 ± 2.22	0.15
	Treated	25.00 ± 8.52		41.00 ± 9.13		50.00 ± 6.14	
IgM	Control	6.81 ± 3.44	0.54	5.00 ± 4.8	0.54	8.07 ± 2.00	0.85
	Treated	11.63 ± 8.45		6.56 ± 6.54		8.00 ± 1.32	

Data indicate mean ± SD. MLN: mesenteric lymph node; JPP: jejunal Peyer’s patch; SPT: soft palate tonsil.

## Data Availability

The data presented in this study are available on request from the corresponding author.

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
