# Peer review of "Effects of Probiotics on Growth and Immunity of Piglets"

_animals, 2022, doi:10.3390/ani12141786_

Round 1
Reviewer 1 Report
Comments to the Authors of manuscript number: animals-1787615 entitled “Effects of probiotics on growth and immunity of the piglets”.
The study was performed on crossbred swine divided to the two groups, control and probiotic supplemented. Body weight was controlled 4 times, finally samples of liver, mesenteric lymph node, jejunal Peyer`s patches and soft palate tonsils were collected to immunological analysis. It is very simple well organized study. The paper is very well prepared. However, I have some questions:
1. Why the blood samples were not collected when pigs were weighed?
2. Why only one dose of probiotic was used?
3. How this dose was determined?
Author Response
We appreciate the time and effort that you dedicated to providing feedback on our manuscript and are grateful for the insightful comments on and valuable improvements to our paper. Our answers to your comments are listed below following each specific question/comment. In the manuscript, the revised portions are shown in red color.
- Why the blood samples were not collected when pigs were weighed?
In this study, we have significant differences phagocytosis and lymphocyte subsets in liver of piglets. We also corrected the blood serum for IgG and IgA concentration on pre and 2 months. There is no significant difference in both groups. We also performed peripheral blood lymphocyte subsets analysis on 2 months. But there is no significant difference in both groups.
- Why only one dose of probiotic was used?
We used probiotics (BIO-THREE Plus) in this study. According to the company’s instruction, the dose of supplementation of probiotics is 0.2-0.5g/head/day. In this study, we used the high dose of probiotics (4g/head/day) for obtain some effects on the probiotics to immune status of liver and secondary lymphoid organ in piglets.
- How this dose was determined?
We used probiotics (BIO-THREE Plus) in this study. According to the company’s instruction, the dose of supplementation of probiotics is 0.2-0.5g/head/day. In this study, we used the high dose of probiotics (4g/head/day) for obtain some effects on the probiotics to immune status of liver and secondary lymphoid organ in piglets.
Reviewer 2 Report
Comments to the authors
§ L63-64: you could describe the aim of your study more clearly
§ L67-69: check the style (point type)
§ L69: correct the ‘’pigpen’’
§ L72-73: rephrase the sentence
§ L85-91: check the style (point type)
§ L106-109: check the style (point type)
§ L111-128: add the appropriate references
§ L135-137: provide details about the records into a separate paragraph in the section of ‘’materials and methods’’
Author Response
We appreciate the time and effort that you dedicated to providing feedback on our manuscript and are grateful for the insightful comments on and valuable improvements to our paper. Our answers to your comments are listed below following each specific question/comment. In the manuscript, the revised portions are shown in red color.
Comments and Suggestions for Authors
Comments to the authors
- L63-64: you could describe the aim of your study more clearly
We changed the aim of this study (L65-66).
- L67-69: check the style (point type)
We checked the style (point type).
- L69: correct the ‘’pigpen’’
We changed (L71-72).
- L72-73: rephrase the sentence
We changed the sentence (L74-79).
- L85-91: check the style (point type)
We checked the style (point type).
- L106-109: check the style (point type)
We checked the style (point type).
- L111-128: add the appropriate references
We added two references and some sentences (L115-116, L269-271, L284-286).
- L135-137: provide details about the records into a separate paragraph in the section of ‘’materials and methods’’
Thank you very much for your suggestion. We changed sentences in L140-141.
Reviewer 3 Report
The purpose of the paper is good, it is ever important to know with depth how the probiotics act.
However, the study used a reduced number of pigs an this is a fragile point of it, specially considering the animal performance that is highlighted as one the most important objective of this study (the title does a mencion about it). Why the authors used this piglets' number, and why they were so different.
Regarding the clearness of the paper, it is OK. The results related to the immunity are well discussed and supported. Nevertheless the results of zoothecnical performance are not shown very well, The Figure 1 is simple and give us doubts about the effectivelly action the porbiotic effect. So, I suggest to show the data in a Table inlcuding their standart deviation or CV. Are there other data about performance, like feed consumption and feed conversion rate? This last parameter is very important.
In terms of discussion the authors just use a simple phrase to explain the action of the probiotics over the performance and this is very general and without references (lines 182 and 183). This should be better discussed. Why the differences just are present in the intermediary age of the piglets and after disappeard.
Considering that the probiotic effect is related with the piglet nutrition, the rations composition and nutritional levels used should be shown, and this information could be supported the discussion too.
After these points, the conclusion can be use the effect on the performance as one effectively virtue of the probiotic in this study. This same comment is valid to the abstract.
Author Response
We appreciate the time and effort that you dedicated to providing feedback on our manuscript and are grateful for the insightful comments on and valuable improvements to our paper. Our answers to your comments are listed below following each specific question/comment. In the manuscript, the revised portions are shown in red color.
Comments and Suggestions for Authors
The purpose of the paper is good, it is ever important to know with depth how the probiotics act.
However, the study used a reduced number of pigs an this is a fragile point of it, specially considering the animal performance that is highlighted as one the most important objective of this study (the title does a mencion about it). Why the authors used this piglets' number, and why they were so different.
We used 17 piglets (seven for controls and ten for probiotics treated). We used adequate space size of the pigpen for this study. We think number difference of piglets in each group is not big concern.
Regarding the clearness of the paper, it is OK. The results related to the immunity are well discussed and supported. Nevertheless the results of zoothecnical performance are not shown very well, The Figure 1 is simple and give us doubts about the effectivelly action the porbiotic effect. So, I suggest to show the data in a Table inlcuding their standart deviation or CV. Are there other data about performance, like feed consumption and feed conversion rate? This last parameter is very important.
Thank you very much for your suggestion. Your suggestion is very important for us. In this study, we did not evaluate feed consumption and feed conversion rate. We added the sentence for limitation in this study in L 190-194.
In terms of discussion the authors just use a simple phrase to explain the action of the probiotics over the performance and this is very general and without references (lines 182 and 183). This should be better discussed. Why the differences just are present in the intermediary age of the piglets and after disappeard.
We added some sentences in discussion (L 187-188, L190-194, L196-199, L220-221, L223-226).
Considering that the probiotic effect is related with the piglet nutrition, the rations composition and nutritional levels used should be shown, and this information could be supported the discussion too.
Thank very much for your suggestion. We did not have the record about detail data about food intake in both groups. We discussed about the limitation in this study and added the sentences in discussion (L190-194).
After these points, the conclusion can be use the effect on the performance as one effectively virtue of the probiotic in this study. This same comment is valid to the abstract. 
Thank very much for your suggestion. We changed the conclusion in L 228-232.